# Differences in Inflammatory Cytokine Profile in Obesity-Associated Asthma: Effects of Weight Loss

**DOI:** 10.3390/jcm11133782

**Published:** 2022-06-29

**Authors:** Marina Bantulà, Valeria Tubita, Jordi Roca-Ferrer, Joaquim Mullol, Antonio Valero, Irina Bobolea, Mariona Pascal, Ana de Hollanda, Josep Vidal, César Picado, Ebymar Arismendi

**Affiliations:** 1Institut d’Investigacions Biomèdiques August Pi i Sunyer (IDIBAPS), 08036 Barcelona, Spain; v.tubita@gmail.com (V.T.); jrocaf@clinic.cat (J.R.-F.); jmullol@clinic.cat (J.M.); valero@clinic.cat (A.V.); bobolea@clinic.cat (I.B.); amdehol@clinic.cat (A.d.H.); jovidal@clinic.cat (J.V.); cpicado@ub.edu (C.P.); earismen@clinic.cat (E.A.); 2Centro de Investigaciones Biomédicas en Red de Enfermedades Respiratorias (CIBERES), 28029 Madrid, Spain; 3Rhinology Unit & Smell Clinic, ENT Department, Hospital Clinic, 08036 Barcelona, Spain; 4Pulmonology and Allergy Department, Hospital Clinic, University of Barcelona, 08036 Barcelona, Spain; 5Immunology Department, CDB, Hospital Clinic, University of Barcelona, 08036 Barcelona, Spain; mpascal@clinic.cat; 6Obesity Unit, Endocrinology and Nutrition Department, Hospital Clínic, 08036 Barcelona, Spain; 7Centro de Investigaciones Biomédicas en Red de Fisopatología de la Obesidad y Nutrición (CIBEROBN), 28029 Madrid, Spain; 8Centro de Investigaciones Biomédicas en Red en Diabetes y Enfermedades Metabólicas (CIBERDEM), 28029 Madrid, Spain

**Keywords:** asthma, bariatric surgery, cytokines, ezrin, obesity, YKL-40

## Abstract

Obesity and asthma are associated with systemic inflammation maintained by mediators released by adipose tissue and lung. This study investigated the inflammatory serum mediator profile in obese subjects (O) (*n* = 35), non-obese asthma (NOA) patients (*n* = 14), obese asthmatics (OA) (*n* = 21) and healthy controls (HC) (*n* = 33). The effect of weight loss after bariatric surgery (BS) was examined in 10 OA and 31 O subjects. We analyzed serum markers including leptin, adiponectin, TGF-β1, TNFR2, MCP-1, ezrin, YKL-40, ST2, IL-5, IL-9, and IL-18. Compared with HC subjects, the O group showed increased levels of leptin, TGF-β1, TNFR2, MCP-1, ezrin, YKL-40, and ST2; the OA group presented increased levels of MCP-1, ezrin, YKL-40, and IL-18, and the NOA group had increased levels of ezrin, YKL-40, IL-5, and IL-18. The higher adiponectin/leptin ratio in NOA with respect to OA subjects was the only significant difference between the two groups. IL-9 was the only cytokine with significantly higher levels in OA with respect to O subjects. TNFR2, ezrin, MCP-1, and IL-18 concentrations significantly decreased in O subjects after BS. O, OA, and NOA showed distinct patterns of systemic inflammation. Leptin and adiponectin are regulated in asthma by obesity-dependent and -independent mechanisms. Combination of asthma and obesity does not result in significant additive effects on circulating cytokine levels.

## 1. Introduction

Obesity (defined as having a body mass index [BMI] > 30 kg/m^2^) is a risk factor for asthma; asthmatic patients with obesity are more likely to have a poor response to corticosteroids (CS) and a severe disease. How obesity affects the pathogenesis and severity of asthma is poorly understood [1,2]. 

Detrimental effects of obesity on lung function have been proposed as a potential mechanism to explain the deleterious effects of obesity on asthma. A restrictive ventilatory alteration has been described in adults with obesity with a predominant decrease in functional residual capacity (FRC) and expiratory reserve capacity (ERC) associated with obstruction in the distal airways [3,4,5]. In children, obesity can contribute towards altering the normal development of the lung with a disproportionate growth of the lung parenchyma with respect to the airways, a phenomenon known as dysanapsis, that results in bronchial obstruction [4,6]. In addition, obesity in adults and children is associated with increased airway hyperresponsiveness (AHR) [4]. A recent study reported that the effects of obesity on pulmonary function in children are transitory and changes from the obstructive to the restrictive pattern in adulthood [7].

There is also a hypothesis that states that the low-grade chronic systemic inflammation present in obesity results in a spillover of pro-inflammatory systemic products from the inflamed adipose tissue, which contributes, via additive or synergistic mechanisms, towards increasing the systemic/airway inflammatory process underlying asthma [1]. Roles for the increased release of some pro-inflammatory cytokines and decreased production of some anti-inflammatory agents have been described [1]. Leptin and adiponectin are the markers most frequently used to assess the relationship between the inflammation of obesity and that of asthma. However, changes in these two markers are difficult to interpret as their systemic levels are influenced by asthma, independently of obesity [8,9,10,11,12,13]. Other makers of systemic inflammation, such as interleukin (IL)-1β, IL-6, IL-8, tumor necrosis factor (TNF)-α, granulocyte-macrophage colony-stimulating factor (GM-CSF), and monocyte chemotactic protein-1 (MCP-1), have also been assessed with some contradictory results [5,13,14]. However, the number of pro- and anti-inflammatory biological products potentially involved in asthma and obesity continuously increases and their potential role in both processes needs to be examined. 

Weight loss after either bariatric surgery (BS) or dietetic treatments associated or not with physical exercise may counteract low-grade systemic inflammation in subjects with obesity [1]. The effect of weight loss on the synthesis of some biological products related to asthma and obesity has been assessed in previous studies in asthmatic subjects with obesity. Concentrations of the pro-inflammatory cytokines IL-6 and IL-8 decreased with weight loss [15], but there was no effect on other cytokines such as IL-8, TNF-α, and GM-CSF [5]. However, the effect of weight loss on some other molecular agents also potentially involved in the inflammatory process in obesity and in obese asthma remains to be examined. 

A deeper understanding of how molecular mechanisms interact between asthma and obesity may help us further understand the complex inflammatory puzzle of asthma associated with obesity. 

The present study aimed to evaluate circulating cytokine levels, before and after BS-induced weight loss, in obese subjects with and without asthma. 

## 2. Materials and Methods

### 2.1. Subjects 

We recruited 35 asthma patients: 21 obese (OA) (BMI ≥ 30 kg/m^2^) and 14 with non-obese asthma (NOA) (BMI < 25 kg/m^2^); 35 obese (O) patients, and 33 age- and gender-matched healthy control (HC) individuals. Forty-one obese (10 OA and 31 O) subjects were evaluated at baseline (V1) and between six and twelve months after they underwent BS (V2). The collaborating endocrinologist presented the objective of the study to the subjects with obesity during the pre-surgery evaluation; those who initially agreed to participate were referred to the Respiratory Department for further studies. 

The following criteria were used to select asthmatic patients: (1) a clinical history of asthma; (2) either bronchodilator responsiveness (>12% and 200 mL improvement in forced expiratory volume in 1 s [FEV_1_] after 180 µg salbutamol metered-dose inhaler) or AHR (PC_20_ methacholine < 8 mg/mL). None of the subjects had received systemic CS for one month or longer prior to evaluation. None of the subjects were current smokers. Forced spirometry was performed according to ERS/ATS standards [16].

### 2.2. Blood Collection and Serum Extraction

Blood samples were obtained by peripheral venipuncture and collected into BD Vacutainer SST™ II Advance tubes. Serum was obtained by centrifugation and then stored at −80 °C until analysis.

### 2.3. Measurement of Serum Cytokine Levels and Immunoglobulin E

Serum leptin, adiponectin, tumor necrosis alpha 2 receptor (TNFR2), MCP-1, ezrin, YKL-40, suppression of tumorigenicity 2 receptor (ST2) for IL-33, IL-5, IL-9, and IL-18 levels were quantified by Luminex^®^ multiplex immunoassay using Human ProcartaPlex Mix&Match kits, and the Human Simplex Kit for transforming growth factor-beta1 (TGF-β1) (Thermo Fisher Scientific, Vienna, Austria) on a Luminex 200 analyzer. Ezrin levels were analyzed in serum using an enzyme-linked immunosorbent assay (ELISA) kit (516351 Cayman chemical). The procedure was carried out in accordance with the manufacturer’s protocol.

IL-17A, IL-6, IL-8, IL-10, TNFR1, IL-13, IL-33, IL-1β, IL-4, interferon gamma (IFN-γ), GM-CSF, and thymic stromal lymphopoietin (TSLP) were also measured. However, in many samples, the levels were below the detection limit and, therefore, could not be accurately assessed. 

Serum levels of total and specific immunoglobulin E (IgE) against common allergens (mites, pollens, cats, dogs, fungi) were measured by immunofluorescence enzyme immunoassay (ImmunoCAP, ThermoFisherScientific, Uppsala, Sweden). Patients with elevated specific IgE against one or more allergens were classified as atopic.

### 2.4. Statistical Analysis

Clinical and experimental data were reported as the median and 25th to 75th percentile. Differences between two groups were analyzed using nonparametric tests: Mann–Whitney U test (unpaired data), Wilcoxon rank test (paired data), or Kruskal–Wallis H test for multiple comparisons. Correlation coefficients were calculated using the Spearman rank method. All analyses were performed using GraphPad Prism version 8.4 for Windows, (GraphPad Software, La Jolla, CA, USA). Statistical significance was defined as *p*-value < 0.05.

## 3. Results

The participants’ demographic and clinical characteristics are shown in Table 1. The level of asthma severity (mild, moderate, severe) was established according to the pharmacological treatment used to control the disease [17,18]. The level of control was assessed using the asthma control test (ACT) [15,16]. OA patients presented reduced forced vital capacity (FVC) and FEV_1_, compared with HC, whereas the FEV_1_/FVC ratio was significantly lower in NOA and OA participants in comparison with O and HC individuals. Blood eosinophilia was higher in NOA subjects, compared with the HC and O groups.

BS was performed in 41 subjects (10 OA and 31 O), in 17 of whom a sleeve gastrectomy was performed (41.5%), while in the remaining 24 (58.5%) a Roux-en-Y gastric bypass was carried out. Table 2 presents the clinical, functional, and inflammatory marker changes after BS. BMI was significantly reduced in both groups of subjects with obesity. Weight loss was associated with a significant improvement in the ACT scores and a reduction in the amount of therapy needed to control the disease. Lung function tests improved in both groups after BS, being statistically significant only in the obese group.

### 3.1. Leptin, Adiponectin, and Adiponectin/Leptin Ratio before and after BS

Circulating levels of pro-inflammatory leptin were significantly higher in O, OA, and NOA than in HC, but there were no differences between them (Figure 1A). Anti-inflammatory adiponectin levels were significantly higher in NOA compared with HC, O, and OA (Figure 1B). Changes in leptin and adiponectin in A, OA, and NOA subjects resulted in a predominant anti-inflammatory adiponectin/leptin ratio in the NOA group relative to the OA and O groups (Figure 1C).

Higher leptin values were observed in non-atopic compared with atopic asthma patients (NOA + OA) [median (25th–75th) 4639 (2824–7425) pg/mL, 2859 (1471–4155) pg/mL, respectively; *p* = 0.041].

Weight loss was associated with a decrease in serum leptin in O subjects, which resulted in a further increase in the adiponectin/leptin ratio (Figure 2B). Serum adiponectin levels did not change after BS. In the OA group, similar results were found, although no statistical significance was reached (Figure 2A). 

### 3.2. Comparison of Serum Inflammatory Markers between HC and the NOA, OA and O Groups

The circulating concentration levels of six molecular products: TGF-β1, TNFR2, MCP-1, ezrin, YKL-40, and ST2 were found to be significantly higher in O subjects with respect to HC. The OA group also showed significantly increased levels of four products compared with HC: MCP-1, ezrin, YKL-40, and IL-18. Finally, in the serum of NOA individuals, four molecular products also had significantly higher concentrations compared with HC: ezrin, YKL-40, IL-5, and IL-18 (Table 3). 

In atopic patients compared with non-atopic patients, we found higher MCP-1 values [median values (25th–75th) 14.5 (10.4–19.7), 9.0 (7.7–15.6) ng/mL, respectively; *p* = 0.0472] and IL-9 values [16.7 (12.0–29.3), 6.1(1.9–13.7), respectively; *p* = 0.0346].

### 3.3. Comparisons between Obese Subjects with and without Asthma

Obese subjects displayed higher serum levels of TGF-β1 and of TNFR2 than their OA counterparts, while IL-9 was the only cytokine showing significantly higher levels in OA with respect to O subjects (Table 4). 

### 3.4. Comparisons between NOA, OA, and HC

A higher adiponectin/leptin ratio was the only statistically significant difference between asthma patients with and without obesity (Figure 1C). IL-5 levels were significantly elevated in NOA patients with respect to HC and O subjects without asthma (Figure 3). Weight loss had no significant effect on IL-5 levels (data not shown).

### 3.5. Correlations

The cytokines MCP-1 (r = 0.368, *p* = 0.0002), TNFR2 (r = 0.295, *p* = 0.0025), YKL-40 (r = 0.391, *p* < 0.0001), ST2 (r = 0.216, *p* = 0.028), ezrin (r = 0.482, *p* < 0.0001), leptin (r = 0.517, *p* < 0.0001), and the adiponectin/leptin ratio (r = −0.241, *p* < 0.0001) correlated with BMI. We further explored the relationship between serum cytokine levels and lung function and found that YKL-40 (r = −0.399, *p* < 0.0001), ezrin (r = −0.229, *p* = 0.02), and leptin (r = −0.350, *p* = 0.0003) presented a negative correlation with FEV_1_ (%). In contrast, the adiponectin/leptin ratio positively correlated (r = 0.253, *p* = 0.01) with FEV_1_ (%). No consistent correlations were observed between blood eosinophilia and serum cytokine levels. Neither were significant differences obtained when asthmatic patients were divided into two groups of more or less than 300 eosinophils/mm^3^.

### 3.6. Impact of Weight Loss on Serum Cytokine Levels 

Weight loss had no statistically significant effect on serum cytokine levels in OA subjects (Figure 4A). However, serum TNFR2, ezrin, MCP-1, and IL-18 levels significantly decreased at the six–twelve-month post-BS follow-up in O subjects (Figure 4B). Other biomarkers in the O group: ST2, YKL-40, IL-9, and TGF-β1 were unaltered (data not shown). 

## 4. Discussion

In the present study, we tested the hypothesis that the spillover of pro-inflammatory substances from adipose tissue will result in enhanced systemic inflammation in obese asthma subjects compared with their non-obese asthma counterparts. The activated systemic inflammation would account for the increased airway inflammation, which in turn would contribute towards explaining the poor response of obese asthmatics to anti-inflammatory therapy. 

The study presents four main findings: (1) Leptin and adiponectin are regulated in asthma by both obesity-dependent and -independent mechanisms, the adiponectin/leptin ratio being the only statistically significant difference between asthma patients with and without obesity; (2) obesity is associated with a systemic inflammation that is more active than that associated with asthma; (3) the combination of asthma and obesity does not result in additive effects on circulating cytokine levels except for IL-9; (4) IL-5 levels were only found to be significantly elevated in NOA patients with respect to HC.

### 4.1. Leptin and Adiponectin in Asthma Are Regulated by Obesity-Dependent and -Independent Mechanisms

Leptin is a protein synthesized and secreted mainly by the adipose tissue [19] and plays a key role in the regulation of appetite and body weight [20]. Leptin promotes cytokine release, generates reactive oxygen species, activates natural killer cells, and up-regulates leukotriene release by alveolar macrophages, multiple actions that collectively promote inflammation [21,22]. Interestingly, serum leptin levels have been found elevated in patients with asthma compared with healthy controls [8,9], an observation suggesting that leptin may contribute to the pathogenesis of the disease by obesity-independent mechanisms. It has been reported that leptin enhances chemotaxis, survival, and secretion of pro-inflammatory cytokines by eosinophils [23,24,25,26,27]. These findings may indicate that an excessive production of leptin may contribute towards activating eosinophilic inflammation and, thereby, increase the severity of eosinophilic asthma [28]. However, the mechanism by which asthma appears to induce leptin synthesis remains to be elucidated. In keeping with previous observations [8,9], our study also found higher leptin levels in asthma patients—particularly in non-atopic patients—and obese subjects than in healthy control individuals. However, no clear additive effects of obesity on leptin levels in asthma were detected. Weight loss was associated with a marked decrease in leptin levels in obese subjects, but it had inconsistent effects in obese asthmatics, possibly reflecting, at least in part, the independent role played by asthma in leptin levels. 

Adiponectin plays a key role in the regulation of insulin sensitivity and energy homeostasis [29]. The role of adiponectin in inflammation is complex and is in part dependent on the tissue wherein the inflammatory process takes place [29]. Along this line, in the endothelium and adipose tissue, adiponectin has anti-inflammatory effects [30,31], while in the synovial, kidney, and intestine, adiponectin exerts pro-inflammatory actions [32,33,34]. The mechanisms underlying these different effects seem complex and include the isoforms of adiponectin involved, the type of macrophages (M1 or M2) activated, and the adiponectin receptor to which it binds. Adiponectin has three different molecular weight isoforms: low (LMW), middle (MMW), and high (HMW). LMW exerts anti-inflammatory effects, while HMW activates pro-inflammatory pathways [29]. In M1 macrophages, adiponectin triggers the expression of pro-inflammatory cytokines (IL-6, TNF-α, IL-12), whereas in M2 macrophages, adiponectin induces the expression of IL-10 [29]. Adiponectin binds to four types of receptors AdipoR1, AdipoR2, T-cadherin, and calreticulin, which are differently distributed among cells and also show different isoform binding affinity. Taken together, all these differences can account for the opposite effects that can be induced by adiponectin [29]. The relationship between asthma, with and without obesity, and serum adiponectin levels has been examined in epidemiological and cross-sectional studies, yielding contradictory results that are difficult to interpret due to the heterogeneity of the studies [29]. Our results show that asthma is associated with increased adiponectin levels, a finding previously published in some [10,11,12] but not all studies [35]. Regarding the inhibitory role of obesity in adiponectin levels in asthma and non-asthmatic subjects, our results partially agree with some works [36,37,38], but not all [39]. The heterogeneity of the study populations (adults vs. children), as well as the differences in BMI and asthma severity of the subjects enrolled in the studies, could account for the observed differences. Childhood-onset obese asthma and obese asthma in adults share some similarities but also show substantial differences [1,4]. Adiponectin levels decrease in a negative relationship to increases in BMI [38,40]. Poor asthma control [39] and asthma severity [41,42] are both associated with lower adiponectin levels. Studies assessing the effects of bariatric surgery found that weight loss was associated with the recovery of serum adiponectin levels in obese [3] and obese asthma [5,14]. Interestingly, the balance of the adiponectin/leptin inflammatory ratio leaned towards normal in asthmatic patients compared with obese asthmatic and non-asthmatic subjects, in whom the ratio tended towards the pattern favoring inflammation. The adiponectin/leptin ratio negatively correlated with BMI. Weight loss was associated with a normalization of the ratio in obese subjects. 

Taken together, all these findings reveal the key role played by adipokines in asthma and their interactions in the regulation of systemic inflammation in obese asthma. 

### 4.2. Obesity Is Associated with a Systemic Inflammation (TGF-β1, TNFR2, MCP-1, Ezrin, YKL-40, and ST2) Which Is More Active Than That Associated with Asthma

TGF-β1 levels increase in parallel with increased adipose body mass tissue, suggesting a role for this cytokine in obesity-related diseases [43,44]. High-fat diet-induced obesity in animal models is associated with increased TGF-β1 signaling expression in the bronchial epithelium resulting in the release of inflammatory mediators and fibrosis [45]. Moreover, serum TGF-β1 levels have been found elevated in asthma [46] and TGF-β1 expression has also been reported as elevated in both structural and inflammatory cells derived from the airways of asthma patients [44,47,48,49]. Induced expression of the TGF-β pathway in the lungs promotes the recruitment of cells such as eosinophils, neutrophils, macrophages, mast cells, and fibroblasts [44,47,50,51,52]. TGF-β1 also induces IL-8 production in human airway smooth muscle cells [53]. Collectively, these observations suggest that TGF-β1 may play an amplifying inflammatory and remodeling role in the airways of obese asthma patients. We found higher elevated serum levels in O subjects, but obesity did not increase serum TGF-β1 levels in OA patients, and in contrast to previous studies in patients with severe asthma [46], we did not find any differences between asthma and healthy control individuals, probably because most of our patients had a well-controlled disease. 

TNF-α is mainly produced by cells of the immune system, such as macrophages, lymphoid cells, and mast cells, and plays relevant regulatory roles in the process of inflammation and healing [54]. Excessive TNF-α production causes systemic inflammation, which is eventually involved in the development of metabolic diseases [54]. TNF-α can bind to two receptors, TNFR1 and TNFR2, which differ in their cellular expression and inflammatory potential [54,55]. TNF-α and TNFRs are expressed as transmembrane proteins that become soluble after being cleaved [55]. Serum TNF-α has been found to be significantly higher in obese subjects compared with healthy individuals [56]. Similarly, serum TNF-α levels have also been reported as elevated in the serum and bronchoalveolar lavage fluid of patients with severe compared with mild asthma and healthy controls [57]. In addition, previous studies have shown that TNF-α signaling via TNFR2 may promote AHR [58]. We found that the serum levels of TNFR2 were significantly higher in obese subjects than in healthy and asthma individuals, with no additive effect between asthma and obesity. The significant and marked decrease in serum TNFR2 levels after BS suggests that the demonstrated beneficial effects of weight loss on AHR [59] may be due, at least in part, to the reduction in the expression of the receptor mainly involved in TNF-α-related AHR. 

IL-18 is a pro-inflammatory cytokine constitutively expressed by various cell types, including monocytes, macrophages, epithelial, and dendritic cells [60]. Immature IL-18 is activated after cleavage by caspase-1 and subsequently released from cells in its bioactive form [61]. The IL-18 receptor (IL-18R) is composed of one binding chain, IL-18Rα, and one signaling chain, IL-18Rβ. [61]. After binding to IL-18, they form a complex that triggers various intracellular pro-inflammatory signals including IRAKs, TRAF6, and NF-κB [62]. In collaboration with IL-12, IL-18 targets CD8+ T-cells and NK cells to stimulate IFN-γ production [63]. In the absence of IL-12, IL-18 promotes Th2 rather than Th1 immune responses and, in combination with IL-23, it activates the Th17 pathway [61]. IL-18 bioactivity is also regulated by the inhibitory effect of IL-18BP [64]. IL-18BP binds mature IL-18 to form high-affinity complexes [64]. In healthy individuals, most of the circulating IL-18 is bound to IL-18BP and, therefore, remains inactivated [65]. Elevated serum levels of free IL-18 have been found in different rheumatological diseases [66]. Previous studies reported that serum IL-18 levels were elevated in obese subjects and were reduced by weight loss [67]. Similarly, high serum IL-18 levels have been found in patients with uncontrolled asthma [68] and in mild and moderate asthma exacerbations [69]. A recent study found that IL-18R was highly expressed in the lung of the most severe asthma patient cluster, supporting that IL-18 signaling may play a key role in the pathogenesis of severe asthma [70]. In line with these observations, we found higher serum IL-18 levels in asthma and obese subjects; however, with no evidence of additive effects. As previously reported [67], weight loss significantly reduced serum IL-18 levels. Collectively, these findings support that the reduction of IL-18 might contribute to the salutary effects of weight loss in asthma. 

Monocyte chemotactic protein-1 (MCP-1) is a member of the CC chemokine families and appears to play a relevant role in asthma pathogenesis because of its ability to attract monocytes and eosinophils, as well as activate mast cells and release leukotriene C-4 into the airway, resulting in AHR [71,72]. Elevated MCP-1 expression has been demonstrated in the bronchial epithelium of asthmatic patients [71], which increases even further preceding asthma exacerbation [73]. Neutralization of MCP-1 significantly inhibits T-cell and eosinophil recruitment to the lung and reduces AHR [74]. One of the most common immune cells infiltrating adipose tissues are macrophages, and their recruitment to adipose tissue is promoted by MCP-1 secreted by the adipose tissue itself [75]. In our study, MCP-1 was found to be significantly higher in obese subjects with and without asthma with respect to healthy controls. Moreover, there was a positive correlation between MCP-1 levels and BMI. A significant decrease in MCP-1 levels was seen with weight loss after BS, a finding in keeping with previous observations [76]. Collectively, these findings support the notion that MCP-1 plays an important role in asthma and obesity pathogenesis. We did not observe any additive effects between asthma and obesity; however, the reduced levels of MCP-1 after BS might partly explain the beneficial effects of weight loss in asthma patients.

IL-33 is a member of the IL-1 family which is expressed by epithelial and endothelial cells as an immature form [77]. When epithelial cells are exposed to irritants, allergens, bacteria, or viruses, IL-33 is cleaved into mature forms via a proteolytic mechanism. Once secreted, IL-33 binds to its specific membrane receptor ST2 (four isoforms) expressed on numerous cells, including immune cells such as innate lymphoid cell type 2 (ILC2), regulatory T cells (Tegs), mast cells, and macrophages [78,79]. Among the four ST2 isoforms, the transmembrane ST2 receptor (ST2L), and the soluble, circulating truncated form of the ST2 protein (sST2) are the best known [78,79]. The soluble form (sST2) is abundantly expressed and secreted by many cell types and tissues, and acts as a decoy receptor, neutralizing free IL-33 in biological fluids [78,79,80,81]. High sST2 levels are found in the serum of patients with inflammatory diseases. When the levels of IL-33 released into the extracellular environment increase, the expression of the soluble sST2 receptor also increases, probably to prevent the severe effects of excessive IL-33 [78,79,80,81]. IL-33 appears to play a major role in the initiation and development of type 2 immune responses involved in the development of asthma. Increased expression of IL-33 has been observed in murine asthma and in human asthma and rhinitis [82,83,84]. Concerning obesity, IL-33 is able to down-regulate excessive inflammation in adipose tissue by targeting immune cells expressing the ST2 receptor. Mice lacking ST2 or IL-33 develop increased adiposity and worsened metabolic profiles. IL-33 treatment contributes to homeostasis in the adipose tissue by facilitating the differentiation and maintenance of Foxp3+ST2+ Treg cells and ILC2 in visceral adipose tissue, an immunomodulatory effect that can contribute towards down-regulating obesity-associated inflammation [79,85]. Because IL-33 quantification in samples of human biological fluids is currently limited by the use of ELISA kits, which lack sensitivity and specificity, in our study we chose to assess the sST2 receptor. We found significantly higher levels of ST2 in obese individuals with respect to healthy controls and asthma subjects. There was a non-significant trend towards increased levels in obese asthmatics, which might suggest an additive effect. sST2 levels significantly correlated with BMI; however, weight loss was not associated with a decrease in sST2 levels. Nevertheless, without data on IL-33 levels, this observation is difficult to interpret as the measurement of sST2 only provides partial information on the balance between IL-33/sST2.

Ezrin is one of the members of the ezrin/radixin/moesin (ERM) family of proteins [86]. ERM proteins exert numerous physiological functions in morphogenesis, cell polarity, and modulation of cell membrane tension [86]. All these varied roles are carried out by ERM proteins by regulating the assembly of protein complexes at the cell surface. Additionally, ezrin, which is localized at the cytoplasmic surface of cellular membranes, links plasma membrane proteins to the cortical actin cytoskeleton [86]. The role of ezrin in the pathogenesis of asthma is, as yet, poorly understood. After allergen exposure, significant increases in serum levels of ezrin were observed in patients with allergic asthma [87]. In contrast to our results, a recent study found that the ezrin levels measured in serum were lower in asthma patients compared with non-asthma controls. In addition, serum ezrin levels were found to be related to the level of asthma control, the worse the control, the lower the ezrin levels [88]. Based on these observations, Jia et al. [88] proposed using serum ezrin levels to monitor asthma control. Most of our asthma patients had a well-controlled disease, a finding that could partly explain the different results between the studies. Despite this relatively good control, ezrin negatively correlated with FEV_1_ in our study, supporting the notion that serum ezrin levels, to some extent, reflect asthma severity. 

To the best of our knowledge, the relationship between obesity and ezrin has not been previously assessed. Interestingly, over-expression of ezrin in pancreatic β-cells results in an enhancement of insulin secretion [89]. Whether ezrin contributes towards overcoming insulin resistance usually in obese subjects is as yet unknown. We found higher levels of ezrin in obese subjects and ezrin levels positively and significantly correlated with BMI. Given the key regulatory role played by the ERM family of proteins, further studies are needed to better understand its role in asthma and obesity. 

Chitin is a biopolymer that is part of the structure of fungi, molluscs, arthropods, and other invertebrates. Chitin is an enzymatic product of chitin synthases and requires specific enzymes—referred to as chitinases—for its degradation. Chitinases probably represent a defensive mechanism against chitin-containing parasites and fungi. Chitinase-3-like protein 1 (CHI3L1), also named YKL-40, is one of several known human chitinases. Elevated serum YKL-40 levels have been found in asthma patients compared with healthy controls [13,90]. Levels of YKL-40 in serum and bronchoalveolar lavage fluid correlated with subepithelial membrane thickness [90], and circulating levels are a significant independent factor associated with annual FEV_1_ decline [91]. Additionally, elevated serum YKL-40 levels have been reported associated with irreversible airway obstruction, poor asthma control, and severe asthma exacerbations [92,93,94,95]. In line with these observations, we found that serum YKL-40 concentrations in NOA were significantly higher than those of HC and, in keeping with previous observations, YKL-40 levels negatively correlated with FEV_1_%, supporting that high serum YKL-40 might be considered a marker of asthma severity [13,93,94]. 

High serum levels of YKL-40 correlating with BMI have also been found in obese children and adults [96,97,98]. The potential additive effects of asthma and obesity on YKL-40 have been evaluated in previous studies with discrepant results. In some studies, OA was associated with significantly higher serum levels compared with NOA, but without differences between NOA and HC [95]. In contrast, other studies found that YKL-40 levels were higher in asthmatics than HC, regardless of obesity [13]. We found the highest YKL-40 levels in OA patients but the difference with respect to NOA was small and non-significant. We did not find any significant effect of weight loss on YKL-40 levels. Similar studies have shown contradictory results, with some studies reporting a decrease in YKL-40 levels [97], while others could not find any significant change [99] after weight loss mediated by the Roux-en-Y gastric bypass. 

### 4.3. IL-9 Was the Only Cytokine Presenting Significantly Higher Levels in OA with Respect to O Subjects

IL-9-producing Th9 cells have been described as a distinct group of CD4+ helper T cells resulting from the combined effects of Interleukin-4 (IL-4) and TGF-β [100,101]. The role of IL-9 in inflammation is a matter of controversy [102,103,104]. On the one hand, IL-9 has been shown to promote Th17 differentiation and production of IL-17 and IL-6 [103]; in contrast, it has also been shown that IL-9 enhances the suppressive function of Treg [104]. Asthma patients have increased numbers of Th9 cells in peripheral blood mononuclear cells (PBMCs) and increased IL-9 levels in both serum and bronchoalveolar lavage fluid (BALF) [105,106,107,108,109]. Th9 cells and IL-9 promote accumulation and activation of T cells, ILC2s, mast cells, and eosinophils and increase levels of IL-5, IL-13, and IgE in animal asthma models [110]. IL-9 concentrations have been found significantly elevated in male obese mice but not in female obese mice with respect to lean mice, suggesting that sexual dimorphism influences IL-9 production, with dominant expression in males [111]. In our study, serum levels of IL-9 were higher in obese asthma patients with respect to healthy controls and obese subjects, a finding that suggests that asthma and obesity might have additive effects on IL-9 secretion. However, there was no significant effect of weight loss on IL-9 serum levels. Since most of our patients were women and sexual dimorphism could influence the effect of obesity on IL-9 levels, further studies in men are needed to better decipher the role of obesity in regulating IL-9 synthesis.

### 4.4. IL-5 Levels Were Only Found to Be Significantly Elevated in NOA Patients with Respect to HC

IL-5 is produced by T helper type 2 cells (Th2), mast cells, and ILC2s, which in turn are activated by epithelium-derivate alarmins, including IL-24, IL-33, and thymic stromal lymphopoietin (TSLP) [112]. Activation, recruitment, and survival of eosinophils are largely driven by IL-5. The contribution of eosinophils to airway inflammation is well recognized; however, results regarding the use of blood eosinophil count as a predictor of asthma exacerbation are controversial and depend upon multiple factors. Indeed, a review published in 2018 showed that elevated blood eosinophil counts were associated with a lower incidence of future acute asthma attacks in patients with asthma that required a previous hospital admission [113]. Indeed, obesity was not associated with a higher risk of asthma hospital admission [114]. However, a recent meta-analysis including 155,772 participants demonstrated blood eosinophil count to be a predictor of asthma exacerbations [115]. Sputum IL-5 levels have been found to be significantly higher in obese asthmatics compared with their lean counterparts. Surprisingly, increased IL-5 levels are not associated with a parallel increase in eosinophils in the sputum, which contrasts with the demonstrated augmentation of eosinophil numbers in the airway submucosa, which appears to be due to an enhanced recruitment of eosinophils to the lung in obese asthmatics [116]. We found higher levels of IL-5 in NOA subjects with respect to HC and O individuals, but we could not find any additive effect of obesity in OA patients. 

### 4.5. Study Limitations

The main limitation of our study was the small size of OA subjected to BS, which significantly reduced the capacity to assess the effects of weight loss on serum cytokine levels in this group of patients. Results need to be examined in a larger cohort. A six to twelve-month follow-up may not be long enough to detect significant changes in some cytokines after BS. Furthermore, we only focused on serum levels, without any parallel information regarding cytokine levels in the airways, where inflammation determines the course and severity of asthma. The impact of asthma inflammatory phenotypes (eosinophilic, neutrophilic, paucigranulocytic) on cytokine levels should also be considered in future studies.

In summary, there has been much speculation and debate regarding the role of obese-related low-grade systemic inflammation as a potentiating factor for systemic and airway inflammation in patients with obese asthma in order to explain why patients with this association frequently present a severe disease, resistant to anti-asthmatic treatment. However, the scientific evidence supporting this hypothesis is very scarce and inconsistent. In our study, we found evidence of partially differentiated systemic inflammation in asthma and obesity. We also observed similar increased levels of some cytokines in subjects with asthma and obesity. However, we could only find additive effects for IL-9 in patients with both diseases. 

## 5. Conclusions

It is likely that the scant progress achieved to date in understanding the mechanisms that associate inflammation in obesity and asthma is due to the great complexity of the mechanisms involved in this association. An example of the complexity is supported by the observation that two adipokines associated with obesity, leptin, and adiponectin, are also up-regulated in asthma via a mechanism that remains to be elucidated. Moreover, the excessive synthesis by the adipose tissue of pro-inflammatory cytokines, such as IL-1β, IL-6, TNF-α, and TGF-β, may contribute to asthma pathogenesis. However, in contrast, enhanced synthesis of pro-inflammatory cytokines in asthma, such as IL-4, IL-5, IL-13, and IL-33, can contribute towards maintaining the lean state [113]. The partial counteracting effects of both inflammations make the interpretation of the complex interplay between asthma and obesity difficult, along with other factors such as differences between OA in children and adults and the influence of different asthma phenotypes on their association with obesity. Finally, weight loss achieved via dietary or surgical therapeutic methods can yield different results in the synthesis of pro-inflammatory mediators. Further studies must take into account all these contributory variables.

## Figures and Tables

**Figure 1 jcm-11-03782-f001:**
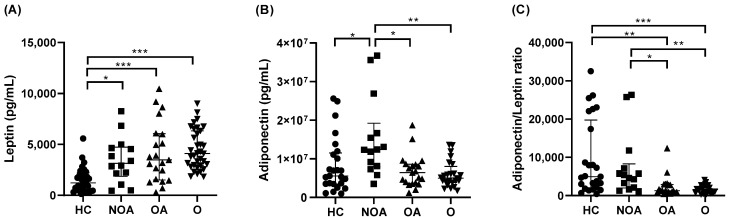
Leptin (**A**), adiponectin (**B**), and adiponectin/leptin ratio (**C**) serum level comparisons between groups. Data presented as individual values and as medians (25th–75th percentile). HC, healthy controls; NOA, non-obese asthmatics; OA, obese asthmatics; O, obese subjects. * *p* < 0.05, ** *p* ≤ 0.01, *** *p* ≤ 0.001; Kruskal–Wallis followed by Dunn’s multiple comparisons test.

**Figure 2 jcm-11-03782-f002:**
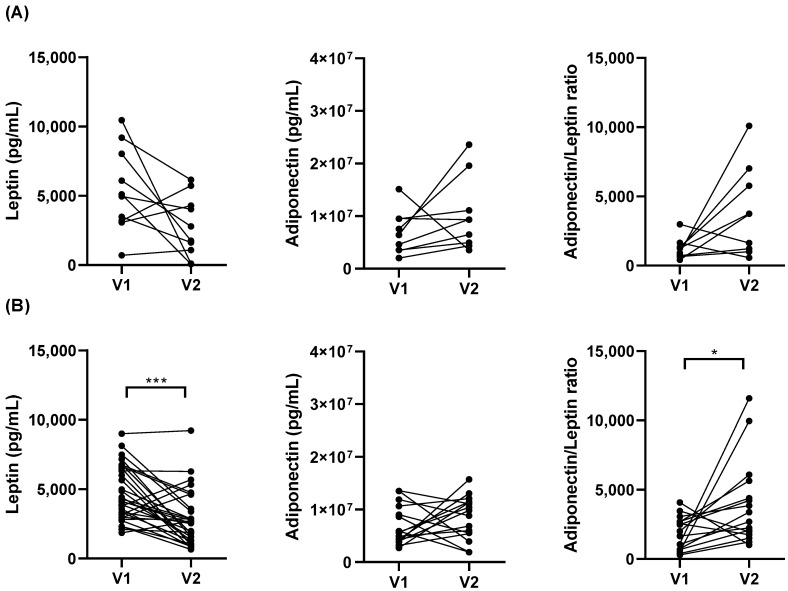
Serum leptin, adiponectin, and adiponectin/leptin ratio values from obese asthmatics (**A**) and obese subjects (**B**) before (V1) and after (V2) bariatric surgery. Data presented as individual values. * *p* < 0.05, *** *p* ≤ 0.001; Wilcoxon test for paired data comparisons between V1 and V2.

**Figure 3 jcm-11-03782-f003:**
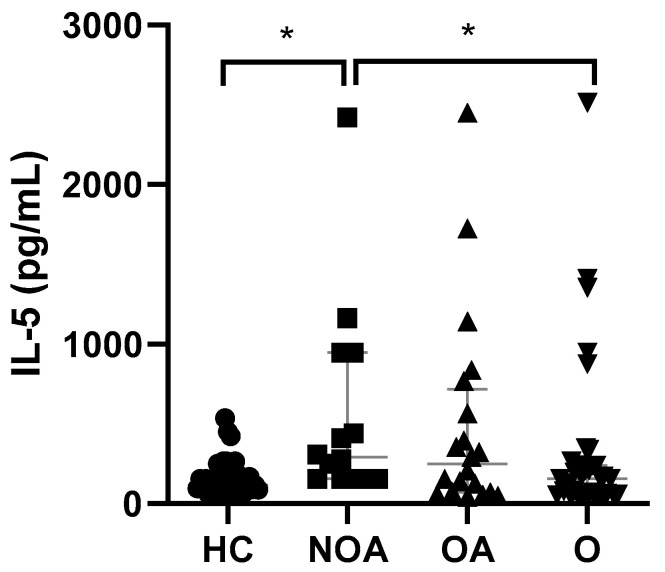
Serum IL-5 level comparisons between groups. Data presented as individual values and as medians (25th–75th percentile). HC, healthy controls; NOA, non-obese asthmatics; OA, obese asthmatics; O, obese subjects. * *p* < 0.05; Kruskal–Wallis followed by Dunn’s multiple comparisons test.

**Figure 4 jcm-11-03782-f004:**
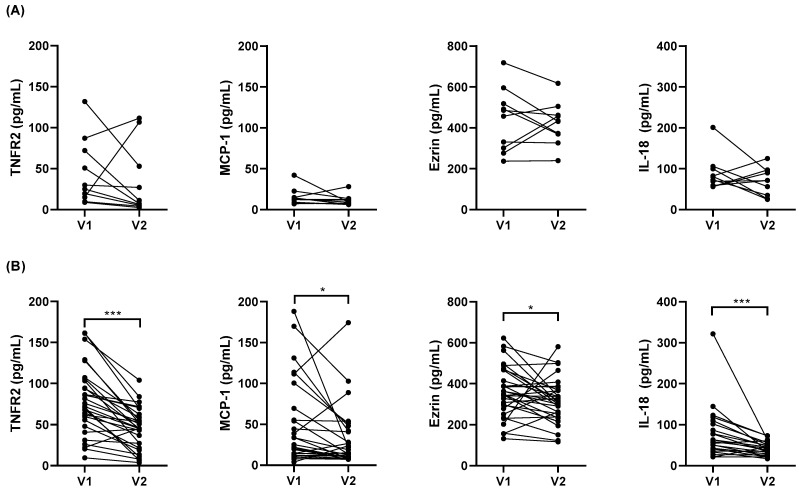
Serum TNFR2, MCP-1, Ezrin, and IL-18 level comparison between asthmatic subjects with obesity (**A**) and obese subjects without asthma (**B**) before (V1) and after (V2) bariatric surgery. Data presented as individual values. * *p* < 0.05, *** *p* ≤ 0.001; Wilcoxon test.

**Table 1 jcm-11-03782-t001:** Baseline demographic and clinical data of the study population.

	HC (*n*, 33)	NOA (*n*, 14)	OA (*n*, 21)	O (*n*, 35)
Age, years	43 (37–50)	52 (43–59)	56 (50–61) *	49 (45–58)
Female, *n* (%)	24 (72)	12 (86)	16 (76)	29 (83)
BMI, kg/m^2^	22.5 (21.6–25.0)	23.2 (22.3–24.9)	38.4 (34.9–44.1) *^,#^	43.4 (39.3–46.9) *^,#^
Mild asthma, *n* (%)	-	0 (0)	4 (19.1)	-
Moderate asthma, *n* (%)	-	6 (42.9)	5 (23.8)	-
Severe asthma, *n* (%)	-	8 (57.1)	12 (57.1)	-
FVC, % pred	126.6 (122.2–136.3)	132.1 (116.7–141.3)	110.4 (94.0–122.5) *^,#^	116.0 (102.4–127.0) *^,#^
FEV_1_, % pred	101.0 (93.5–109.0)	86.5 (75.2–103.8)	85.0 (61.0–95.5) *	90.5 (84.0–101.0)
FEV_1_/FVC	79.0 (76.0–83.0)	71.0 (57.0–75.0) *	77.0 (69.5–82.0)	82.0 (79.0–84.0) ^#,†^
ICS ^§^, *n* (%)	-	13 (92.9)	19 (82.6)	-
Atopia, *n* (%)	-	8 (57.1)	10 (43.5)	-
IgE, kU/L	26.9 (16.5–80.2)	95.5 (34.8–306.0)	63.7 (14.4–273.0)	48.3 (18.9–110.0)
Eosinophils, %	2.0 (1.3–3.2) ^#^	4.7 (3.6–6.3)	3.3 (2.3–5.0)	2.4 (1.6–3.2) ^#^
Eosinophils, cells/mm^3^	100 (100–200)	300 (200–425) *	200 (125–300) *	200 (100–200) ^#^
Neutrophils, %	60.5 (55.2–65.6)	54.8 (51.4–61.0)	56.7 (51.6–64.6)	63.2 (56.4–66.2) ^#^
Neutrophils, cells/mm^3^	3.5 (3.0–4.3)	3.6 (3.2–4.2)	3.7 (3.0–4.9)	4.5 (3.7–5.1) *

Data presented as medians (25th–75th percentile). HC, healthy controls; NOA, non-obese asthmatics; OA, obese asthmatics; O, obese subjects; BMI, body mass index; FVC, forced vital capacity; pred, predicted; FEV_1_, forced expiratory volume in 1 s; ICS, inhaled corticosteroids; IgE, immunoglobulin E. * *p* < 0.05, compared with HC; ^#^ *p* < 0.05, compared with NOA; ^†^ *p* < 0.05 compared with OA; Kruskal–Wallis followed by Dunn’s multiple comparisons test. ^§^ For NOA and OA patients who received ICS, the mean ± SD of the ICS dose in budesonide equivalents was 557.1 ± 256.6 and 1222.1 ± 862.9 µg/day, respectively.

**Table 2 jcm-11-03782-t002:** Demographic and clinical data of asthmatic and non-asthmatic obese patients before (V1) and six to twelve-months after (V2) bariatric surgery.

	Obese Asthmatics (*n*, 10)		Obese (*n*, 31)	
	V1	V2	V1	V2
Age, years	53 (47–58)	54 (49–59)	49 (45–59)	50 (47–59)
Female, *n* (%)	10 (100)	10 (100)	26 (84)	26 (84)
BMI, kg/m^2^	44.1 (38.7–47.1)	29.9 (26.4–34.8) *	42.7 (38.7–45.9)	29.0 (27.4–30.8) ^#^
Mild asthma, *n* (%)	4 (40)	9 (90)	-	-
Moderate asthma, *n* (%)	4 (40)	1 (10)	-	-
Severe asthma, *n* (%)	2 (20)	0 (0)	-	-
ACT	18 (18–24)	25 (25–25) *	-	-
FVC, % pred	110.6 (96.9–121.5)	119.0 (111.9–136.3)	116.5 (104.3–129.2)	130.4 (116.7–141.7) ^#^
FEV_1_, % pred	89.0 (79.5–97.2)	96.5 (92.5–104.8)	93.5 (84.7–101.0)	102.0 (89.0–109.0) ^#^
FEV_1_/FVC	80.5 (76.5–82.5)	81.5 (77.5–83.0)	82.0 (79.0–84.0)	81.0 (75.0–83.0) ^#^
ICS ^§^, *n* (%)	6 (60)	2 (20)	-	-
IgE, kU/L	35.4 (17.3–117.5)	35.7 (20.3–64.0)	51.7 (16.8–115.0)	28.3 (14.4–62.8)
Eosinophils, %	2.9 (1.9–3.7)	2.5 (1.4–3.3)	2.4 (1.6–3.4)	2.2 (1.6–2.8)
Eosinophils, cells/mm^3^	200 (100–225)	200 (75–225)	200 (100–200)	100 (100–200) ^#^
Neutrophils, %	56.7 (53.5–64.3)	56.2 (54.7–58.5)	62.5 (55.5–65.3)	56.9 (52.3–64.4)
Neutrophils, cells/mm^3^	3.6 (2.9–4.4)	3.3 (2.9–4.2)	4.4 (3.7–5.1)	3.1 (2.6–4.2) ^#^

Data presented as medians (25th–75th percentile). BMI, body mass index; ACT, asthma control test; FVC, forced vital capacity; pred, predicted; FEV_1_, forced expiratory volume in 1 s; ICS, inhaled corticosteroids; IgE, immunoglobulin E. * *p* < 0.05, compared between V1 and V2 in the obese asthmatic group, ^#^ *p* < 0.05 compared between V1 and V2 in the obese group; Wilcoxon test. ^§^ For OA patients before and after bariatric surgery who received ICS, the mean ± SD of the ICS dose in budesonide equivalents was 643.3 ± 496.1 and 150.0 ± 70.7 µg/day, respectively.

**Table 3 jcm-11-03782-t003:** Serum cytokine level comparison of non-obese asthmatics, obese asthmatics, and obese subjects with healthy controls.

		HC	NOA	OA	O	HC vs. NOA	HC vs. OA	HC vs. O
						*p*-Value	*p*-Value	*p*-Value
TGF-β1	N,	33	14	21	35			
	ng/mL	361.2	349.1	348.6	495.4	0.9725	0.7382	0.0155
	25th–75th	(158.2–516.4)	(255.1–442.0)	(239.2–416.1)	(288.0–746.0)			
TNFR2	N,	32	14	21	34			
	ng/mL	32.4	45.5	45.1	70.0	0.0792	0.1163	<0.0001
	25th–75th	(8.8–56.2)	(25.0–99.2)	(22.4–65.8)	(45.8–96.7)			
MCP-1	N,	28	14	21	34			
	ng/mL	8.6	9.7	14.5	21.3	0.5388	0.0441	0.0003
	25th–75th	(6.5–17.8)	(8.1–14.7)	(10.6–28.1)	(11.5–75.7)			
Ezrin	N,	33	14	21	35			
	ng/mL	216.9	325.5	322.5	341.7	0.0025	0.0005	<0.0001
	25th–75th	(160.4–285.4)	(217.2–361.0)	(232.1–471.1)	(251.1–414.0)			
YKL-40	N,	33	14	21	34			
	ng/mL	23,440	58,428	60,240	59,151	<0.0001	<0.0001	<0.0001
	25th–75th	(13,720–39,958)	(47,346–103,219)	(44,494–171,396)	(41,590–86,112)			
ST2	N,	32	14	21	35			
	ng/mL	999.1	931.2	1009.0	1570.0	0.6667	0.7903	0.0238
	25th–75th	(676.5–1456.0)	(576.0–1308.0)	(486.2–2229.0)	(820.8–3207.0)			
IL-5	N,	33	14	20	35			
	ng/mL	116.6	294.2	252.1	155.5	0.0004	0.1443	0.7442
	25th–75th	(93.6–191.1)	(155.5–949.6)	(72.3–718.2)	(58.1–240.7)			
IL-9	N,	28	8	14	28			
	ng/mL	3.2	4.9	12.8	3.4	0.3530	0.0171	0.7666
	25th–75th	(2.0–7.5)	(1.8–16.4)	(2.2–19.3)	(1.9–7.1)			
IL-18	N,	27	14	21	32			
	ng/mL	40.7	57.8	69.3	56.9	0.0207	0.0118	0.0648
	25th–75th	(22.3–81.0)	(30.4–152.6)	(40.9–102.9)	(34.0–99.4)			

Data presented as medians (25th–75th). HC, healthy controls; NOA, non-obese asthmatics; OA, obese asthmatics; O, obese subjects. Mann–Whitney U test.

**Table 4 jcm-11-03782-t004:** Serum cytokine level comparison between obese groups with and without asthma.

		OA	O	OA vs. O
				*p*-Value
TGF-β1	N,	21	35	
	ng/mL	348.6	495.4	0.0103
	25th–75th	(239.2–416.1)	(288.0–746.0)	
TNFR2	N,	21	34	
	ng/mL	45.1	70.0	0.018
	25th–75th	(22.4–65.8)	(45.8–96.7)	
MCP-1	N,	21	34	
	ng/mL	14.5	21.3	0.1528
	25th–75th	(10.6–28.1)	(11.5–75.7)	
Ezrin	N,	21	35	
	ng/mL	322.5	341.7	0.8273
	25th–75th	(232.1–471.1)	(251.1–414.0)	
YKL-40	N,	21	34	
	ng/mL	60,240	59,151	0.6002
	25th–75th	(44,494–171,396)	(41,590–86,112)	
ST2	N,	21	35	
	ng/mL	1009.0	1570.0	0.1502
	25th–75th	(486.2–2229.0)	(820.8–3207.0)	
IL-5	N,	20	35	
	ng/mL	252.1	155.5	0.221
	25th–75th	(72.3–718.2)	(58.1–240.7)	
IL-9	N,	14	28	
	ng/mL	12.8	3.4	0.0299
	25th–75th	(2.2–19.3)	(1.9–7.1)	
IL-18	N,	21	32	
	ng/mL	69.3	56.9	0.3049
	25th–75th	(40.9–102.9)	(34.0–99.4)	

Data presented as medians (25th–75th percentile). OA, obese asthmatics; O, obese subjects. Mann–Whitney U test.

## Data Availability

Data generated or analyzed during this study are included in this published article or available from the corresponding author on request. All analyzed datasets were sourced from the authors.

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
