# Peer review of "Differences in Inflammatory Cytokine Profile in Obesity-Associated Asthma: Effects of Weight Loss"

_jcm, 2022, doi:10.3390/jcm11133782_

Round 1
Reviewer 1 Report
This study investigated the inflammatory cytokine profile is serum samples obtained from subjects in four groups: obese subjects, non-obese asthmatics, obese asthmatics and healthy controls. In addition, some effects of bariatric surgery was evaluated in the two obese groups. The authors conclude that the combination of asthma and obesity does not result in additive effects on cytokine levels in serum. Furthermore, they state that the interpretation of the complex interplay between asthma and obesity is difficult and requires further studies.
Overall the manuscript reads well for me as a non-english native speaker. The authors aimed to evaluate circulating cytokine levels, before and after weight loss after bariatric surgery, in obese subjects with and without asthma. However, a major concern is that most of data is presented from the obese group and not the obese asthma group. The methods used in the study seems adequate. However, methods, results and figures needs some clarification. The discussion is quite long and the many references point to the fact that most of the data have previously been shown by others. If any novel data is presented, or any new conclusions can be drawn by the data presented, the message is not conveyed clearly.
Some detailed comments can be found below.
2.1 Subjects and Table 1 and 2: I’m missing some information regarding the description of the subject included in the study that I believe would benefit the reader.
· The authors write that none of the subjects received systemic CS, but were they on inhaled CS or other treatment?
· Atopy status and how it was determined (sIgE or skin prick test)?
· Smoking status: current smoker, ex-smoker or never smoker.
· The authors have included eosinophils in percent but also mention dividing the subjects based on eos/mm3. Both values would be informative.
· What about other blood leukocytes such as neutrophils, monocytes, lymphocytes and basophils? Would be interesting, both in regards to asthma and inflammation related to obesity.
· In the legend of table 1 and table 2 it’s written Vitamin D but no values can be found, neither is it mentioned in the result. In table 2 legend Adipo/Lept ratio values are missing. Please explain how the ratio Adipo/Lept is calculated.
Line 148: Lung function tests improved after BS. This only occurred in the obese group but not the obese asthma group and should be stated.
Figure 2: There seems to be figures missing here. Line 172 refer to figure 2C. Figure legend mention all four groups but I assume it is only obese subjects in 2a and 2b?
Line 180: What do the authors mean by “After excluding leptin and adiponectin, ….”?
Line 212: The sentence refers to figure 2A?
Line 213: Is really the IL-5 levels elevated in O subjects compared to NOA subject, refers to figure 4?
Figure 3: This figure seems to show the same data as Table 3.
Table 4: Table heading and table says obese asthmatics and obese subjects but table legend says non-obese asthmatics and obese asthmatics. Which comparison are we looking at?
Line 272: Data of leptin levels in obese asthmatics has not been shown.
Line 328: Data of reduced TGF-B1 levels in O subjects after weight loss is missing.
Line 502: The authors state that the main limitation was the low number of OA subjects subjected to bariatric surgery and weight loss. If I understand correctly this is the reason for not presenting any cytokine data. Even though a larger cohort would have been preferred, data from a small cohort could be interesting especially since it was the key cohort.
Reviewer 2 Report
This very interesting paper provides relevant results regarding the circulating cytokine levels, before and after BS-induced weight loss, in obese subjects with and without asthma. The authors observed that O, OA, and NOA showed distinct patterns of systemic inflammation. Leptin and adiponectin are regulated in asthma by obesity-dependent and -independent mechanisms. Combination of asthma and obesity does not result in significant additive effects on circulating cytokine levels.
The manuscript is well organized. Methods are appropriate, results are clearly described and illustrated, as well as properly discussed. References are relevant and updated. Therefore, this paper requires minor corrections and can be very useful for Journal of Clinical Medicine readers, because it provides very interesting information within the current context of published studies.
Reviewer 3 Report
The present article entitled ”Differences in Inflammatory Cytokine Profile in Obesity-associated Asthma: Effects of Weight Loss” address the differences in inflammatory cytokine profile in patients with asthma and obesity, as well as the effect of weight loss in this profile. The authors have included 4 different groups of subjects, including healthy controls (HC), obese subjects (O), and patients with asthma and obesity (OA) and without obesity (NOA), in which they measure many different parameters related to systemic inflammation: leptin, adiponectin, TGF-β1, TNFR2, MCP-1, ezrin, 27 YKL-40, ST2, IL-5, IL-9, IL-18. Obesity is a very common cofounding factor in asthma and can also participate in the pathogenesis of this pathology. Therefore, a deep understanding on the changes specifically associated to obesity in asthma could help to understand their relationship and be important for treatment purposes. The authors clearly explain all their results in the discussion, so I just have some comments and suggestions that authors have to addressed before accepting for publication:
- Line 102. Please add the tubes you used for serum collection.
- Lines 143-148. Table 2. Did you performed a Chi^2 test to assess change in proportion of mild-moderate-severe? It seems this clearly change with the surgery and would be nice to include statistic behind and mentioned on text.
- Figures 1, 3, 4 and Table 3. Statistical test used (Mann-Whitney U test) is wrong, as it can only be used when for compare only two groups. Authors should use a test for multiple comparisons (e.g., Kruskal-Wallis test) followed by posthoc test (e.g., Dunn´s test). Please change this, and change results and discussion accordingly in case found different significances.
- It would be interesting to classify asthma patients in atopic and non-atopic, as atopy could clearly influence the levels of inflammatory mediators measured. I recommend to include this data in the table and a figure could be potentially included comparing those two groups at least in supplementary or at least mention in text. I can imagine obese asthma are mainly non-atopic. Maybe interesting to compare OA non atopic with NOA non-atopic.
Discussion:
Authors have performed a deep discussion on all aspects evaluated in the article. I only have minor suggestions, as it is important to mention both sides in some of the questions that are controverse in literature:
- Line 488-500. Authors could not find an increase in IL-5 in OA compared to NOA. Indeed, results around the use of blood eosinophil count as predictors of asthma exacerbation are controverse and depend upon multiple factors. Indeed, a review made in 2018 (PMID: 29514744) have shown that elevated blood eosinophil counts are associated with a lower incidence of future acute asthma attacks in patients with asthma that required a previous hospital admission. Indeed, obesity was not associated to a higher risk of asthma hospital admission (PMID: 31947560). However, a recent meta-analysis including 155,772 participants (PMID: 33135257) demonstrated BEC is a predictor of asthma exacerbations. This should be also discussed in this paragraph.
